# Using Explainabilty to Detect Adversarial Attacks

## Abstract

Deep learning models are often sensitive to adversarial attacks, where carefully-designed input samples can cause the system to produce incorrect decisions. Here we focus on the problem of detecting attacks, rather than robust classification, since detecting that an attack occurs may be even more important than avoiding misclassification. We build on advances in explainability, where activity-map-like explanations are used to justify and validate decisions, by highlighting features that are involved with a classification decision. The key observation is that it is hard to create explanations for incorrect decisions.

We propose EXAID, a novel attack-detection approach, which uses model explainability to identify images whose explanations are inconsistent with the predicted class. Specifically, we use SHAP, which uses Shapley values in the space of the input image, to identify which input features contribute to a class decision. Interestingly, this approach does not require to modify the attacked model, and it can be applied without modelling a specific attack. It can therefore be applied successfully to detect unfamiliar attacks, that were unknown at the time the detection model was designed. We evaluate EXAID on two benchmark datasets CIFAR-10 and SVHN, and against three leading attack techniques, FGSM, PGD and C&W. We find that EXAID improves over the SoTA detection methods by a large margin across a wide range of noise levels, improving detection from $\sim 70\%$ to over 90% for small perturbations.

## 1 Introduction

Machine learning systems can be tricked to make incorrect decisions, when presented with samples that were slightly perturbed, but in special, adversarial ways (Szegedy et al., 2013). This sensitivity, by now widely studied, can hurt networks regardless of the application domain, and can be applied without knowledge of the model (Papernot et al., 2017). Detecting such adversarial attacks is currently a key problem in machine learning.

To motivate our approach, consider how most conferences decide on which papers get accepted for publication. Human classifiers, known as reviewers, make classification decisions, but unfortunately these are notoriously noisy. To verify that their decision are sensible, reviewers are also asked to explain and justify their decision. Then, a second classifier, known as an area-chair or an editor, examines the classification, together with the explanation and the paper itself, to verify that the explanation supports the decision. If the justification is not valid, the review may be discounted or ignored.

In this paper, we build on a similar intuition: Explaining a decision can reduce misclassification. Clearly, the analogy is not perfect, since unlike human reviewers, for deep models we do not have trustworthy methods to provide high level semantic explanation of decisions. Instead, we study below the effect of using the wider concept of explanation on detecting incorrect decisions, and in particular given adversarial samples that are designed to confuse a classifier. The key idea is that different classes have different explaining features, and that by probing explanations, one can detect classification decisions that are inconsistent with the explanation. For example, if an image is classified as a dog,

but has an explanation that gives high weight to a striped pattern, it is more likely that the classification is incorrect.

We focus here on the problem of *detecting* adversarial samples, rather than developing a system that provides *robust* classifications under adversarial attacks. This is because in many cases we are interested to detect that an attack occurs, even if we cannot automatically correct the decision.

The key idea in detecting adversarial attacks, is to identify cases where the network behaves differently than when presented with untainted inputs, and previous methods focused on various different aspects of the network to recognize such different behaviours Lee et al. (2018); Ma et al. (2018); Liang et al. (2018); Roth et al. (2019); Dong et al. (2019); Katzir & Elovici (2018); Xu et al. (2017). To detect these differences, here we build on recent work in explainability Lundberg & Lee (2017b). The key intuition is that explainability algorithms are designed to point to input features that are the reason for making a decision. Even though leading explainability methods are still mostly based on high-order correlations and not necessarily identify purely causal features, they often yield features that people identify as causal (Lundberg & Lee, 2017a). Explainability therefore operates directly against the aim of adversarial methods, which perturb images in directions that are not causal for a class. The result is that detection methods based on explainability holds the promise to work particularly well with adversarial perturbations that lead to nonsensical classification decisions.

There is second major reason why using explainable features for adversarial detection is promising. Explainable features are designed to explain the classification decision of a classifier trained on non-modified (normal) data. As a result, they are independent of any specific adversarial attack. Some previous methods are based on learning the statistical abnormalities of the added perturbation. This makes them sensitive to the specific perturbation characteristics, which change from one attack method to another, or with change of hyperparameters. Instead, explainability models can be agnostic of the particular perturbation method.

The challenge in detecting adversarial attacks becomes more severe when the perturbations of the input samples are small. Techniques like C&W Carlini & Wagner (2017b) can adaptively select the noise level for a given input, to reach the smallest perturbation that causes incorrect classification. It is therefore particularly important to design detection methods that can operate in the regime of small perturbations. Explanation-based detection is inherently less sensitive to the magnitude of the perturbation, because it focuses on those input features that explain a decision for a given class.

In this paper we describe an EXAID (EXplAIn-then-Detect), an explanation-based method to detect adversarial attacks. It is designed to capture low-noise perturbations from unknown attacks, by building an explanation model per-class that can be trained without access to any adversarial samples.

Our novel contributions are as follows: We describe a new approach to detect adversarial attacks using explainability techniques. We study the effect of negative sampling techniques to train such detectors. We also study the robustness of this approach in the regime of low-noise (small perturbations). Finally, we show that the new detection provides state-of-the-art defense against the three leading attacks (FGSM, PGD, CW) both for known attacks and in the setting of detecting unfamiliar attacks.

## 2 Related work

### Explainable AI.

Several methods have been recently proposed to address black box decisions of AI systems. LIME (Ribeiro et al., 2016) is based on locally approximating the model around a given prediction with a simple interpretable model (e.g. a decision tree). DeepLIFT (Shrikumar et al., 2017) uses a modified version of back propagation to compute the contribution of each input feature to the output. SHAP (Lundberg & Lee, 2017a) approximates

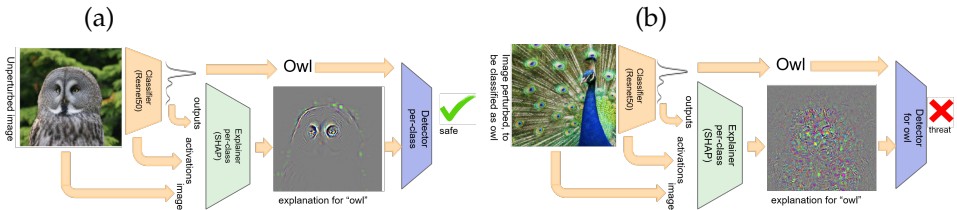

Figure 1: **Illustration of EXAID**. First, an image is classified by a standard image classification system like ResNet. Then, an explanation is created based on the image, the network activations and the network output. Finally, a detector checks if the generated explanation is consistent with the predicted label. (a) An image of an owl is correctly classified, and the produced explanation is consistent with the label "owl". (b) An image of peacock is perturbed and used as an attack. It is falsely classified as an owl, and is detected as adversarial because its explanation is inconsistent with the predicted label.

the Shapley values of input features, which were derived in cooperative game theory to distribute the total gains to the players. Specifically, for explaining deep neural networks, SHAP use a variant of DeepLIFT as a approximation for Shapley values.

## Adversarial attacks

The literature on adversarial attacks is vast. We focus here on three high-performing adversarial attacks which are relevant for the experiments. Each of the three represents a group of attacks that share the same main idea.

**Fast Gradient Sign Method (FGSM).** This attack by (Goodfellow et al., 2014) creates a perturbation by "moving" an example one step in the direction of the gradient. Let $c$ be the true class of $x$ and $J(C, x, c)$ be the loss function used to train our deep neural network $C$. The perturbation is computed as a sign of the model's loss function gradient $\Delta x = \epsilon * sign(\nabla_x J(C, x, c))$, where $\epsilon$ ranging from 0.0 to 1.0. The parameter $\epsilon$ controls the magnitude perturbation and can be thought as the noise-level of the adversarial sample.

**Projected Gradient Decent (PGD).** Madry et al. (2017) suggested to improve FGSM, in the following way. One can interpret FGSM as a one-step scheme for maximizing the inner part of the saddle point formulation. A more powerful adversary will be a multi-step variant, which essentially applies projected gradient descent on the negative loss function $x^{t+1} = x^t + \epsilon * sign(\nabla_x J(C, x, c))$ while $x^0 = x$.

**Carlini and Wagner (C&W).** Carlini & Wagner (2017b) employed an optimization algorithm to seek the smallest perturbation that enables an adversarial example to fool the classifier. As showed in (Carlini & Wagner, 2017a), this attack is considered to be one of the most powerful attacks, and therefore is a common baseline.

When designing attacks, previous studies took into account various factors: the probability that the attack is successful, the effect on the appearance of a perturbed image, and the time it takes to run the attack. The above three methods prioritize these aspects differently, reaching different tradeoff operating points. Specifically, FGSM is usually faster and the C&W attack yields less-visible perturbation of the input images.

## Detecting adversarial attacks

Several previous techniques have been proposed to detect adversarial examples. Liang et al. (2018) measured the effect of quantization and smoothing of the image on the network classification, both parameterized as a function of image entropy. Similarly, Xu et al. (2017) suggested to reduce the degrees of freedom of the input space by applying transformations like quantization and smoothing, and then compare model predication before and after each transformation. Ma et al. (2018) measures the characteristics of the region surrounding a reference example. Lee et al. (2018) models the distribution of ac-

tivation at the hidden layers of the classifier, using a Gaussian mixture model, and feeds the likelihood to a classifier. Katzir & Elovici (2018) models the changes in the labels of a k-nn for each activation layer in the base model. Pang et al. (2018) propose using a new loss in training, which encourages the neural network to learn latent representations that better distinguish adversarial examples from normal ones. Roth et al. (2019) models the statistical robustness of log-odds to perturbations, for normal and adversarial examples. Generally speaking, these methods assume that adversarial examples differ intrinsically from natural images, either in the sample space or because the perturbation affects propagation of activity in the neural network. Some of those methods require modifying the base model. Very recently, (Fidel et al., 2019) described an explanation-based approach to detection, related to the current paper.

## 3 EXAID: EXPLAIN THEN DETECT

EXAID consists of two components: (1) **Explain**. Create per-class explanations for both correct predictions and incorrect ones. (2) **Detect**. Train a binary classifier to decide if an explanation is consistent with the class decision. These two components are schematically shown in Figure 1.

### 3.1 EXPLAIN

The first step in EXAID implements an explanation model. Given a pretrained classifier that may be attacked, we used an explainability model to extract explanations for every sample classified by the model. The explanation model can take as input the raw input image, as well as the whole base model architecture and weights, and produces an explanation in the terms of the input features. Formally it is a function that maps a sample and a classifier, and its prediction into explanation space $\mathcal{E} : (x, f_{theta}(x)) \to R^n$, where $f_{theta}$ is a classification model producing a predicted label $y = output(f(x))$.

Since our goal is to learn which explanations are typical for each class, we collected both *positive explanations* - applying an explanation model to a correct prediction of the network, and *negative explanations* - corresponding to incorrect predictions of the model.

#### SAMPLING EXPLANATIONS

Creating positive explanations $\mathcal{E}(x_i, y_i)$ is usually straight forward, as one simply applies the explanation model on each sample that was correctly classified $f(x_i) = y_i$. More care should be given to creating negative explanations. We consider three types of negative explanations: **wrong negatives**, **adversarial negatives** and **other-class negatives**.

First, one may collect samples $(x_i, y_i)$ where the model made an incorrect decision $f(x_i) \neq y_i$, and collect their explanations $\mathcal{E}(x_i, f(y_i))$. We name these **wrong negatives**. For models that are well trained, the number of these explanations is small. Furthermore, not all classes are confused by other classes, and only some classes may lead to explanations of some other classes.

Second, one can employ an adversarial attack on the training data and collect negative explanations of adversarially perturbed samples. We name them **adversarial negatives**. As with wrong negatives, these explanations correspond to cases where the model made an incorrect decision, but unlike wrong negatives the explanations may have a different distribution, because the input was designed to confuse the network. Even if the specific type of adversarial attack is not known, these samples may be useful because they are based on fooled decisions and may reflect typical patterns of adversarial examples. However, training against an incorrect attack may cause overfitting to a specific type of attack and hurt detection accuracy.

Third, for every labeled sample $(x_i, y_i)$, we produce explanations $\mathcal{E}(x_i, y)$ for all incorrect classes $y \in \mathcal{Y}, y \neq y_i$. For example, for a car image correctly classified as a car, we produce explanations for classes like dogs and cats. These explanations are used as **other-class negatives** for the correct class $y_i$.

### The explainability model

As an explainable AI approach we used SHAP deep explainer. As shown in (Lundberg & Lee, 2017b) SHAP is considered a leading explainer, providing explanations that have stronger agreement with human explanations than other methods. We therefore believe it is likely to capture the "correct" features by which people make labeling decisions. In addition, Lundberg & Lee (2017b) has shown that SHAP is the only explainer that has both local accuracy and consistency, which are desirable properties.

### 3.2 Detect

Given a set of positive and negative explanations per class, we train a deep binary classifier per class, to detect explanations that are inconsistent with model predictions. Note that in this settings, it is natural to train a detector as a binary multiclass multi-label classifier, and not as a multiclass classifier, because we wish to condition the decision on the prediction of the image classifier.

When training the detector, one may consider two learning setups, aiming to protect against *unknown-attacks*, or against *familiar attacks*. It appears as if defending against a known attack would be an easier task, because one may learn the properties of the attack. Unfortunately, since new attacks can be easily designed, it is highly desirable to devise generic defenses.

We address this topic by controlling the data that is used for training the detector. Specifically, we consider two variants of EXAID.

***EXAID familiar***. During training, the binary detector is presented with adversarial negatives. It can therefore learn a distribution of explanations resulting from a specific adversarial attack. Specifically, we trained using high-noise FGSM.

***EXAID unknown***. The binary detector is not presented with any adversarial negatives during training. The only negative explanation the classifier trained on are other-class negatives and wrong negatives.

Below we tested both variants on the known attack (FGSM) and on unfamiliar attacks (PGD, C&W).

## 4 Experimental results

We evaluated EXAID on two benchmark data sets, in the task of attack detection. Our code will be available at `https://github.com/[anonymous-author]/EXAID`.

### Datasets

We evaluated EXAID on two standard benchmarks: CIFAR10 (Krizhevsky et al.) and SVHN (Netzer et al., 2011). As Carlini & Wagner (2017a) showed, MNIST is not a good dataset for evaluating adversarial defences. This is probably due to the fact that it is a low-dimension dataset, making it easier to detect changes an attacker made to the image. (Carlini & Wagner, 2017a) show their results on CIFAR-10. In order to show the validity of our results on more than one dataset, we also used SVHN that has similar complexity. The CIFAR-10 dataset consists of 60,000 32x32 colour images in 10 classes, with 6,000 images per class. There are 50,000 training images and 10,000 test images. The SVHN dataset is obtained from house numbers in Google Street View images. It consists of more than 600,000 32x32 colour images in 10 classes. While similar in flavor to MNIST, it comes from a significantly more diverse distribution. We used the 73,257 digits provided for training and the 26,032 digits for testing.

IMPLEMENTATION DETAILS

For both CIFAR-10 and SVHN we used a pretrained Resnet34 as a base model. To train the EXAID detector we extract positive explanation, wrong negative and other-class negative from natural images as described in algorithm 1. The *EXAID-unknown* model was trained on those explanations. To train *EXAID-familiar* we extracted adversarial negative using a FGSM attack with a noise level of $\epsilon = 0.1$.

---

**Algorithm 1** Create positive and negative explanation, and train bad explanation detector for each class

---

1: **Input:** $F$ - Trained model, $(X, Y)$ - Dataset of labeled samples
2: **Initialize:** positives $\leftarrow$ array of empty sets, negatives $\leftarrow$ array of empty sets
3: **for** $(x, y) \in (X, Y)$ **do**
4:     $\hat{y} = F(x)$
5:     $explanation = SHAP(F, x)$
6:     **if** $\hat{y} == y$ **then**
7:         $positives[\hat{y}].append(explanation[\hat{y}])$
8:         **for** $i = 1..\hat{y} - 1, \hat{y} + 1..n$ **do**
9:             $negatives[i].append(explanation[i])$ ; // Collect other-class negative
10:         **end for**
11:     **else**
12:         $negatives[\hat{y}].append(explanation[\hat{y}])$ ;     // Collect wrong negatives
13:     **end if**
14: **end for**
15: **for** $i = 1..n$ **do**
16:     $C_i \leftarrow$ Train(positives[i],negatives[i]) ;    // Train classifier for class i
17: **end for**

---

As described, we used SHAP as an explainability model (Lundberg & Lee, 2017a). The original implementation of SHAP runs on CPU and is therefore suitable for use only on a small number of samples, rather than on entire datasets. To enable us to run SHAP on a large number of examples we modified the original implementation to run on GPU. The new version is available in our Git repository.

DEFENSE BASELINES

We compared EXAID with three recently-published adversarial detection baselines, and two new variants of these baselines.

**(1) ANR** (Liang et al., 2018). A method based on measuring the effect of quantization and smoothing of the image on the network classification, both are parameterized as a function of the image entropy. We used the implementation provided by the authors. Since ANR was not tested in the original paper on CIFAR-10 and SVHN as done here, we tuned the hyperparameters of their method using hyperopt (Bergstra et al., 2013).

**(2) Mahanalobis** (Lee et al., 2018). This approach models the distribution of activations in the hidden layers of the classifier, as obtained in response to natural (unperturbed) samples, using a Gaussian mixture model. Given a set of likelihood scores from the GMM, a classifier is trained to determine if a set of activations is obtained in response to an adversarial example or a natural one. That classifier is trained on adversarial examples. We used the implementation provided by the authors, and as the original paper, we trained the classifier with adversarial examples crafted by FGSM.

**(3) Mahanalobis Unsupervised**. We modified the method of (Lee et al., 2018) to reach an attack-agnostic baseline as follows. Instead of training an attack-dependent discriminator on adversarial samples, we estimated the likelihood of a set of network activations as the product of likelihoods of all layers.

**(4) LID** (Ma et al., 2018). LID measures the characteristics of the region surrounding a reference example, and give it a likelihood score. This is done separately for each representation of the example, in the classifier's hidden layers. As in Mahalanobis, a classifier

is trained to determine if a set of activations is obtained in response to an adversarial example or a natural one. We used the implementation from (Lee et al., 2018), and trained the classifier with adversarial examples crafted by FGSM.

**(5) Unsupervised LID**. As for Mahalnobis, we test an unsupervised version of LID, based on the product of likelihoods of individual layers, without training a classifier.

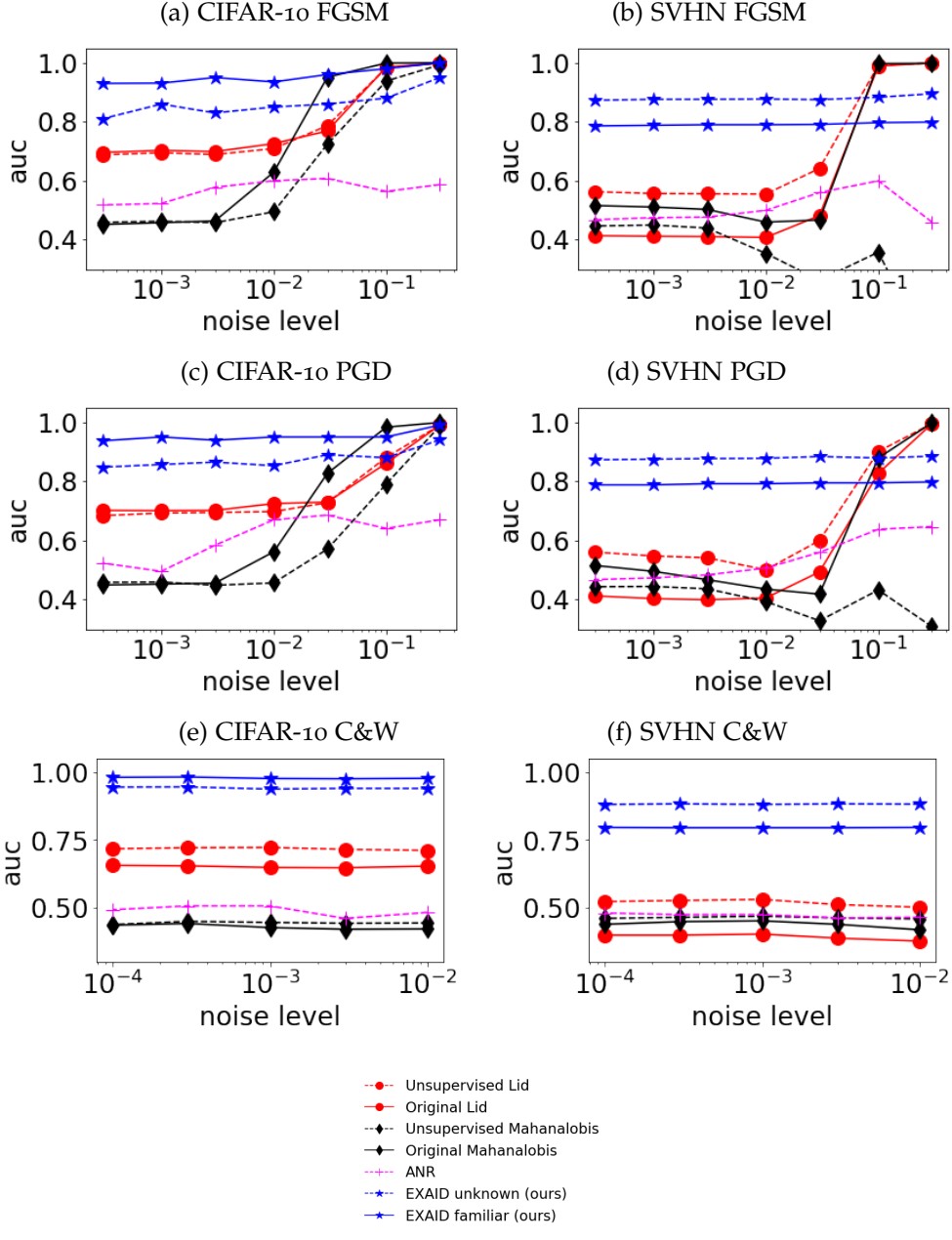

Figure 2: **Defense methods comparison**. Each sub-figure compares 2 EXAID variants to five baselines. (a,b) defend against FGSM, for CIFAR-10 and SVHN. (c,d) same against PGD (d,e) Same against C&W. EXAID outperforms all baselines in low-noise scenarios, and is comparable in the high-noise regime.

### 4.1 EXPERIMENTAL SETUP

We test the detection models against *oblivious adversaries*, an attack scenario in which an attacker has full knowledge about the model (white box attack), but is not aware of the existence of the defense model. We believe this is a relevant scenario, since in the real world, most attackers will not have direct access to the attacked model and its defense. In this case, the attacker will be forced to use a black box attack. However, as (Papernot et al., 2017) showed, adversarial examples are transferable between models. Given transferability, attacking a black box model is not marginally harder than a white box. Because of that, we baseline our model against a white box attack as which is harder to detect. This is not the case when the model is defended, since (Li et al., 2019) shows that the transferability of adversarial examples works well between vanilla neural networks, but fail to transfer between defended neural networks.

We believe that the magnitude of perturbation used by an attack is a major factor that determines the success of adversarial detection methods. There is still no clear protocol in the literature about comparing attacks and detections depending on this factor, and different reported experiments use different values. We therefore repeated all experiments for a wide range of noise levels and report performance across that wide range.

### 4.2 ATTACKS

We used three attack methods to test EXAID: (1) One step gradient attack (FGSM) (Goodfellow et al., 2014), (2) Iterative projected gradient (PGD) (Madry et al., 2017) and the Carlini and Wagner attack, which uses optimization to add as small as possible perturbation (C&W) (Carlini & Wagner, 2017b). All attacks were implemented using Advertorch (Ding et al., 2019). Opposed to other defense methods benchmarks, we examined the effect of noise-level on a range of three orders of magnitude.

## 5 RESULTS

The results for all detection methods are shown in figure 2. EXAID significantly outperforms the other methods when the noise level is small (small perturbations), and with attack methods that use adaptive noise levels (C&W). Typically, the AUC is increased from 70% to over 90%.

LID and Mahalanobis both perform well in high noise scenarios, and slightly outperform EXAID on SVHN in these scenarios. However, when the noise level decreases LID and Mahalanobis performance suffers drastically, while EXAID's remains high. Interestingly, our unsupervised variant of LID, performs at least as well, and sometimes better, than the original LID. This may be because LID was trained with FGSM samples and may deteriorate in cross-attack scenarios. These findings show the importance of benchmarking defense models against a wide range of noise levels.

## 6 CONCLUSION

In this paper we proposed EXAID, a novel attack-detection approach, which uses model explainability to identify images whose explanations are inconsistent with the predicted class. Our method outperforms previous state-of-the-art methods, for three attack methods, and many noise-levels. We demonstrated that the attack noise level has a major impact on previous defense methods. We hope this will encourage the research community to evaluate future defense methods on a large range of noise-levels.

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
