# OpenReview forum: "Using Explainabilty to Detect Adversarial Attacks"
_ICLR.cc/2020/Conference — Reject_

### Official Review · AnonReviewer1 · 2019-10-23
**Official Blind Review #1**

**Rating:** 3

**Review:**

This paper suggests a method for detecting adversarial attacks known as EXAID, which leverages deep learning explainability techniques to detect adversarial examples. The method works by looking at the prediction made by the classifier as well as the output of the explainability method, and labelling the input as an adversarial example if the predicted class is inconsistent with the model explanation. EXAID uses Shapley values as the explanation technique, and is shown to successfully detect many standard first-order attacks.

Though method is well-presented and the evaluation is substantial, the threat model of the oblivious adversary is unconvincing. The paper makes the argument that oblivious adversaries are more prevalent in the real world, but several works [1,2,3,etc.] have shown that with only query access to input-label pairs from a deep learning-based system, it is possible to construct black-box adversarial attacks. Thus, it is unclear why an attacker cannot just treat the detection mechanism as part of this black box, and mount a successful query-based attack.

Though I recognize that the task of detection is separate from the task of robust classification, in both cases the defender should at least operate in the case where the attacker has input-output access to the end-to-end system (including whatever detection mechanisms are present). In particular, it seems impossible to "hide" a detector from an end user (when the method detects an adversarial example, it will alert the user somehow that the input was rejected), and so the user will be able to use this information to fool the system. The authors should investigate the white-box accuracy of their detection system, or at the very least try black-box attacks against the detector. For this reason I do not recommend acceptance for the paper at this time.

[1] https://arxiv.org/abs/1804.08598
[2] https://arxiv.org/abs/1807.04457
[3] https://arxiv.org/abs/1712.04248

**Experience Assessment:**

I have published in this field for several years.

**Review Assessment: Checking Correctness Of Derivations And Theory:**

N/A

**Review Assessment: Checking Correctness Of Experiments:**

I carefully checked the experiments.

**Review Assessment: Thoroughness In Paper Reading:**

I read the paper thoroughly.

---

### Official Review · AnonReviewer3 · 2019-10-24
**Official Blind Review #3**

**Rating:** 3

**Review:**

A Simple method to detect adversarial examples, but needs more work.

#Summary:
The paper proposed a method that utilizes the model’s explainability to detect adversarial images whose explanations that are not consistent with the predicted class.  The explainability is generated by SHAP, which uses Shapley values to identify relative contributions of each input to a class decision. It designs two detection methods: EXAID familiar, which is aimed to detect the known attacks and EXAID unknown, which is against unknown attacks. Both of the two methods are evaluated on perturbed test data which are generated by FGSM, PGD and CW attack with perturbations of different magnitudes. Qualitative results also show that the proposed method can effectively detect adversaries, especially when the perturbation is relatively small.

#Strength
The method is easy to implement and using the idea of interpretation for detecting adversarial examples seems interesting.

Good results are demonstrated compared with other comparators.

#Weakness
The idea of this paper is based on the interpretation method of DNN. However, it has been shown that these interpretation methods are not reliable and easy to be manipulated [1][2]. Therefore, although the method is simple to design, it also brings other security concerns.
Unfortunately, the paper does not address these issues. In addition, the comparators listed in the experiments are not state-of-art or common baselines. It is either not clear why authors modified the existing method and develop their own “unsupervised” version.
In the experiments, many details are omitted. For example, how is the “noise level” defined? Are they based on L1, L2 or L-inf perturbation? For PGD attack, how many iterations does the generation run and what is the step size? How many effective adversarial examples are generated for training and testing? And all the experiments are conducted in a relatively small dataset, it is also suggested to do experiments on large datasets, e.g. Imagenet.
In the evaluation part, it looks strange to me why the EXAID familiar performs worse than EXAID unknown in evaluating FGSM attack on SVHN since the EXAID familiar is trained using FGSM attack.

#Presentation
I think the authors used a wrong template to generate the article. The font looks strange and the headnote indicates it is prepared for ICLR2020. The paper contains many typos and even the title contains a misspelling. Poor coverage of citations. There are more works for detecting adversarial examples that are published, e.g. [3][4][5]. On the other hand, the paper does not have the literature review for work related to the model interpretation.

Overall, I think the paper is not good enough for publication at ICLR.
[1] Dombrowski, Ann-Kathrin, et al. "Explanations can be manipulated and geometry is to blame." arXiv preprint arXiv:1906.07983 (2019).
[2] Ghorbani, Amirata, Abubakar Abid, and James Zou. "Interpretation of neural networks is fragile." Proceedings of the AAAI Conference on Artificial Intelligence. Vol. 33. 2019.
[3] Meng, Dongyu, and Hao Chen. "Magnet: a two-pronged defense against adversarial examples." In Proceedings of the 2017 ACM SIGSAC Conference on Computer and Communications Security, pp. 135-147. ACM, 2017.
[4] Liao, Fangzhou, Ming Liang, Yinpeng Dong, Tianyu Pang, Xiaolin Hu, and Jun Zhu. "Defense against adversarial attacks using high-level representation guided denoiser." In Proceedings of the IEEE Conference on Computer Vision and Pattern Recognition, pp. 1778-1787. 2018.
[5] Ma, Shiqing, Yingqi Liu, Guanhong Tao, Wen-Chuan Lee, and Xiangyu Zhang. "NIC: Detecting Adversarial Samples with Neural Network Invariant Checking." In NDSS. 2019.


**Experience Assessment:**

I have read many papers in this area.

**Review Assessment: Checking Correctness Of Derivations And Theory:**

N/A

**Review Assessment: Checking Correctness Of Experiments:**

I carefully checked the experiments.

**Review Assessment: Thoroughness In Paper Reading:**

I read the paper thoroughly.

---

### Official Review · AnonReviewer4 · 2019-10-27
**Official Blind Review #4**

**Rating:** 1

**Review:**

Summary: The authors propose an explanation-based adversarial example detection algorithm. The main idea is to train a discriminator to detect whether the explanatory saliency map is consistent with the input. Experiments have been conducted on CIFAR10 and SVHN to validate the method.

Comments:
+ The idea is straightforward and easy to follow.

- The use of SHAP as the only explanation method is not well explained. There are a plenty of works on visual explanation methods, such as guided-backprop[1], excitation-backprop[2], integrated gradient[3], Grad-CAM[4], real-time saliency[5] and so on. And based on my expertise, SHAP cannot generate the most accurate saliency among these methods.  If the proposed framework is general, why not to conduct ablation study on the different choice of explainer?

- Doubts on the effectiveness of the proposed method. According to former works[6, 7], explanatory saliency methods  are vulnerable and unreliable with respect to input perturbations. But in this paper, the authors assume that the explanation saliency map for normal examples are perfectly correct and used as positive instances for training the discriminator. I think they only focus on target attack, in which the attacking target label is semantically distinct from the original label, and the resulting saliency map distribution is very different from the correct one. However, considering a tabby cat image is perturbed to become tiger cat, since two classes are very close, the resulting saliency maps should be similar and the detector may fail to detect the adversarial example. Therefore, I encourage the authors to provide more results on this challenging scenario (for example, conduct un-target attack on imagenet dataset).

- The reported results in Figure 2(e) is abnormal. First, the blue line (authors' method) is very close to AUC=1.0 across different noise levels, which means that the detector can perfectly classify all the adversarial examples in all the situation. Second, the reported values of other methods are not correct. For example, the black line (original Mahalanobis) is below AUC=0.5 across all the noise level. However, in the Table3 ResNet-CIFAR10 row of its original paper[8], the reported AUC under C&W attack is 95.84, which is much larger than those shown in the figure. Therefore, I think the comparison is invalid. Similar problems also appear in Figure 2(f).

[1] J. Springenberg, A. Dosovitskiy, T. Brox, and M. Riedmiller. (2015). Striving for simplicity: The all convolutional net. In ICLR (workshop track).
[2] J. Zhang, Z. Lin, J. Brandt, X. Shen, and S. Sclaroff. (2016). Top-down neural attention by excitation backprop. In ECCV.
[3] Sundararajan, M., Taly, A., & Yan, Q. (2017). Axiomatic attribution for deep networks. In ICML.
[4] Selvaraju, R. R., Cogswell, M., Das, A., Vedantam, R., Parikh, D., & Batra, D. (2017). Grad-cam: Visual explanations from deep networks via gradient-based localization. In ICCV.
[5] Dabkowski, P., & Gal, Y. (2017). Real time image saliency for black box classifiers. In NeurIPS.
[6] Kindermans, P. J., Hooker, S., Adebayo, J., Alber, M., Schütt, K. T., Dähne, S., ... & Kim, B. (2019). The (un) reliability of saliency methods. In Explainable AI: Interpreting, Explaining and Visualizing Deep Learning (pp. 267-280). Springer, Cham.
[7] Adebayo, J., Gilmer, J., Muelly, M., Goodfellow, I., Hardt, M., & Kim, B. (2018). Sanity checks for saliency maps. In NeurIPS
[8] Lee, K., Lee, K., Lee, H., & Shin, J. (2018). A simple unified framework for detecting out-of-distribution samples and adversarial attacks. In NeurIPS

**Experience Assessment:**

I have published one or two papers in this area.

**Review Assessment: Checking Correctness Of Derivations And Theory:**

I carefully checked the derivations and theory.

**Review Assessment: Checking Correctness Of Experiments:**

I carefully checked the experiments.

**Review Assessment: Thoroughness In Paper Reading:**

I read the paper thoroughly.

---

### Official Review · AnonReviewer2 · 2019-10-27
**Official Blind Review #1995**

**Rating:** 3

**Review:**

The paper proposes a method to check whether a model is under attack by using state of the art explainability model, SHAP. They evaluated their technique using CIFAR-10 and SVHN w.r.t. 5 baseline techniques. They showed their method outperforms all the other baselines with a significant margin.

+ Overall I think the paper made a valuable contribution to the adversarial ML literature. Using explainability to detect the presence of adversarial attacks seems like a nice intuitive idea and the results show that it indeed works.

-	However, the contribution of the paper is rather incremental. They just used SHAPE to adversarial and negative examples. I do not see any insight while explaining the results.

-	Why under PGD and FGSM attack under higher noise, the proposed technique is similar or slightly worse than Lid and Mohalonabis baselines?

-	I would also like to see how these results hold good for a complicated dataset like ImageNet






**Experience Assessment:**

I have read many papers in this area.

**Review Assessment: Checking Correctness Of Derivations And Theory:**

N/A

**Review Assessment: Checking Correctness Of Experiments:**

I assessed the sensibility of the experiments.

**Review Assessment: Thoroughness In Paper Reading:**

I read the paper at least twice and used my best judgement in assessing the paper.

---

### Public Comment · ~Anthony_Wittmer1 · 2019-09-27
**Interesting work, how about evalating on Nattack?**

Hi, this paper is an interesting work. However, I have some questions about the evalation.

I think a stronger attack is missing in the evalation, i.e., Nattack[1]. Since the baseline LID (Ma et al., 2018). has been broken by Nattck with the attack success rate of 100%, I have the question whether the proposed method provides the true robustness against the adversarial examples.  That it, I am wondering does the proposed method suffer from the same attack?

It would be solid to include further experiments on the robustness against Nattack in the paper.

[1] NATTACK: Learning the Distributions of Adversarial Examples for an Improved Black-Box Attack on Deep Neural Networks. ICML 2019

---

> ### Author Response · Authors · 2019-10-20
> **EXAID also detects Nattack, outperforms baselines.**
>
> Thank you for your important feedback and helpful suggestion!
>
> We originally discussed Nattack [1] when explaining the attack scenario, but did not compare with it directly to keep the focus on attack detection, rather than model robustness.
>
> Following your comment, we further evaluated our detection approach with the [1] attacks. Specifically, we used the implementation provided by the authors for attacking LID (github.com/Cold-Winter/Nattack/tree/master/lid) using their best published hyper parameters, and made minor adjustments to fit our pytorch model. We also reduced the population size from 300 to 200 so the attack model fit our K40 GPU RAM. We evaluated our defense using the successful adversarial images.
>
> For Nattack,  EXAID (our approach) again consistently outperforms other detection baselines, on both CIFAR and SVHN, while keeping detection rates at the same ball park as with other attacks. Specifically, on CIFAR, EXAID improves detection AUC over the baselines, from 0.70 (ANR), 0.68 (unsupervised LID), 0.67 (original LID), 0.43 (unsupervised Mahalanobis) and 0.46 (original Mahalanobis) to *0.96* (EXAID familiar) and 0.89 (EXAID unknown).
>
> Similarly, on SVHN, EXAID improves from 0.53 (ANR), 0.63 (expand LID), 0.49 (original LID), 0.35 (expand Mahalanobis) and 0.56 (original Mahalanobis), to *0.95* (EXAID unknown) and 0.78 (EXAID familiar).
>
> We will add detailed results to the next version of the paper.

---

> > ### Public Comment · ~Anthony_Wittmer1 · 2019-10-21
> > **Strange results**
> >
> > Sorry, I find the result about Nattack in terms of LID is strange and unconvincing.
> >
> > As the reported result by the work of Nattack , Nattck has broken the detection of LID with the attack success rate of 100%. That is, the result of LID  on Nattack is 0%.
> >
> > However, as the reply shown, the result of LID  on Nattack reported by the authors is 67%, which is close with the clean accuracy (66.9%) reported by the work of Nattack and has a big gap with the previous result (0%). Maybe the minor adjustments make something wrong for Nattack.

---

> > > ### Author Response · Authors · 2019-10-23
> > > **Attack detection and Model robustness are different tasks**
> > >
> > > Thank you for your comment.
> > >
> > > There is a fundamental distinction that should be stressed between two different tasks:  (A) Build robustness against an attack, and (B) detect that an attack was made. While the tasks are related, they are fundamentally different.
> > >
> > > The current paper discusses attack detection. The question points out that the results differ from a model-robustness paper, which is expected.
> > >
> > > More specifically: Consistent with previous papers, we do find that running Nattack on our model, the success rate of the attack was 100% on CIFAR-10. However, the current paper aims to *detect* successful adversarial examples rather than make a model more robust.  Also, we did not use Nattack to attack the robust LID model that was used in the Nattack paper (which has accuracy of 66.9%), but to attack our base model, which is unprotected, and has accuracy of 87%. We used our own model instead of the robust model to maintain consistency across the rest of the experiments. In this setup, which was justified in the paper, the attack was not aimed to evade LID detection, so it isn't surprising Nattack didn't completely evade LID detector.

---

### Decision · Program_Chairs · 2019-12-19

**Decision:**

Reject

**Comment:**

This paper proposes EXAID, a method to detect adversarial attacks by building on the advances in explainability (particularly SHAP), where activity-map-like explanations are used to justify and validate decisions. Though it may have some valuable ideas, the execution is not satisfying, with various issues raised in comments. No rebuttal was provided.